# ReaKE: Contrastive Molecular Representation Learning with Chemical Synthesis Knowledge Graph

## Abstract

Molecular representation learning has demonstrated great promise in bridging machine learning and chemical science and in supporting novel chemical discoveries. State-of-the-art methods mostly employ graph neural networks (GNNs) with self-supervised learning (SSL) and extra chemical reaction knowledge to empower the learned embeddings. However, prior works ignore three major issues in modeling reaction data, that is ***abnormal energy flow***, ***ambiguous embeddings***, and ***sparse embedding space*** problems. To alleviate these problems, we propose ReaKE, a chemical synthesis knowledge graph-driven pre-training framework for molecular representation learning. We first construct a large-scale chemical synthesis knowledge graph comprising reactants, products and reaction rules. We then propose triplet-level and graph-level contrastive learning strategies to jointly optimize the knowledge graph and molecular embeddings. Representations learned by ReaKE can capture the changes between the before and after of a reaction (template information) without prior information. Extensive experiments of downstream tasks and visualization demonstrate the effectiveness of our method compared with the state-of-art methods.

## 1 Introduction

Organic chemistry is rapidly developed with the growing interest in big data technology(Schwaller et al., 2021b). Among them, reaction prediction becomes a necessary component of retro-synthesis analysis or virtual library generation for drug design(Kayala & Baldi, 2011). However, the prediction of chemical reaction outcomes in terms of products, yields[1], or reaction rates with computational approaches remains a formidable undertaking. In the last years, natural language processing (NLP)-based methods showed robustness and effectiveness in representing molecules and reaction prediction(Schwaller et al., 2020), these methods treat the precursors' Simplified molecular-input line-entry system (SMILES)[2] as text. While effective, they face the challenges of dealing with molecules' structural information.

To handle this challenge, researchers leverage the ascendency of Graph neural networks (GNNs) in modeling 2D molecular structures(Liu et al., 2019; Yang et al., 2019; Liu et al., 2022; Ma et al., 2022). Still, there exists the problem of predicting out-of-distribution data samples since labeled data are limited and the chemical space is complex(Wu et al., 2018). Thus, some recent methods employ self-supervised learning (SSL)strategies to use unlabelled data, including designing special pretext tasks and applying the contrastive learning framework(You et al., 2020; Zhang et al., 2020; Xu et al., 2021; Wang et al., 2022; Li et al., 2022). However, SSL on molecular graph structures remains challenging as the current approaches mostly lack domain knowledge in chemical synthesis. Recent studies have pointed out that pre-trained GNNs with random node/edge masking gives limited improvements and often lead to negative transfer on downstream tasks Hu et al. (2020); Stärk et al. (2021), as the perturbations actions on graph structures can hurt the structural inductive

---

[1]Reaction yield is a measure of the quantity of moles of a product formed in relation to the reactant consumed, obtained in a chemical reaction, usually expressed as a percentage.

[2]It is a specification for unambiguously describing molecular structures in ASCII strings. For example, the SMILES string of ethanol is 'CCO'.

bias of molecules. More recently, a few studies inject extra chemical reaction knowledge into SSL training to empower the learned embeddings(Wen et al., 2022; Wang et al., 2021). Among them, the state-of-art method MolR(Wang et al., 2021) preserves the equivalence of molecules with respect to chemical reactions in the embedding space. (i.e., forcing the sum of reactant embeddings and the sum of product embeddings to be equal for each chemical equation.)[3]

Albeit promising, the chemical reaction-aware method face the following three problems:

**(1) *Abnormal energy flow***: all chemical reactions are accompanied by changes in entropy, and changes in entropy require reaction conditions such as temperature and pressure to trigger. Under the equivalence assumption of the previous method, the reactants and products can flow with each other as long as the embedding is equal, which violates the principle of entropy increase in the second law of thermodynamics. For example, given a reaction $A + B \rightarrow C$ and a reaction of $D + E \rightarrow C$, it will result in $A + B \rightarrow D + E$, but that reaction might not occur.

**(2) *Ambiguous embeddings***: the previous method assumes that the embeddings of reactants and products are equal in embedding space, however, reactants and products are often similar but totally different in property, this assumption will lead to a lack of discrimination between reactants and products in the embedding space, for example, incorrectly predicting products as reactants, more detailed examples are in Table 5 of Appendix F (such as No.5 and No.72 reactions).

**(3) *Sparse embedding space***: since the amount of recorded chemical reactions is limited, the embedding spaces of reactants and products learned by the previous methods are sparse and lack smoothness, which may lead to a large offset of embeddings when making a small perturbation to the reaction. i.e., if I have an $A \rightarrow B$ reaction, there will be $e(A) = e(B)$, $e(\cdot)$ represents the embedding function. Suppose a small perturbation $\sigma$ (removing an atom outside the reaction center) is done on both $A$ and $B$, it may cause a large offset due to the sparsity of the chemical space and make $e(A + \sigma) \neq e(B + \sigma)$.

To address these problems, we develop ReaKE, a novel deep learning framework that learns chemistry-meaningful molecular representations from graph-in-graph data architecture, i.e., a knowledge graph (KG) that connects 2D molecular graphs using reaction templates. First, to alleviate the *energy flow* and the *ambiguous embedding* problems, we construct a chemical synthesis knowledge graph and build explicit connections between molecules through *reaction template* information. This can introduce the changes in reaction sites as the trigger conditions of flow between molecules, but also help distinguish reactants and products in the embedding space. Then, for solving the *sparse embedding space* problem, we further design a functional group-based SSL method for reaction triplet-level representation learning, which can help build a denser chemical embedding space. Finally, we propose a reaction-aware contrastive learning strategy to improve the efficiency of the knowledge graph-level training.

Extensive experiments demonstrate that the representations learned by our proposed model can benefit a wide range of downstream tasks which require chemical synthesis priors information. For example, ReaKE achieves a 6.8% absolute Hit@1 gain in pretext reaction prediction, an average of 9.4% absolute $F_1$ score gain in reaction classifications, and an average 4% $R^2$ improvement in yield predictions over existing state-of-the-art methods, respectively. Further visualization studies indicate that our reaction representations can not only categorize reactions clearly but also capture discriminative properties of reaction templates.

## 2 METHODS

An illustrative overview of our proposed method of molecular pre-training with **Rea**ction **K**nowledge **E**mbedding (**ReaKE**) is presented in Fig. 1. In this section, we first introduce the definition of a chemical synthesis knowledge graph (section 2.1), as schematically shown in Fig. 2(a). Then we depict the joint learning of the triplet-level encoder and the knowledge encoder at the graph-level (section 2.2), followed by the overall pre-training objects (section 2.3).

Knowledge graph embedding (KGE) aims to encode components of a KG into a low-dimensional continuous vector space to support the downstream graph operations and knowledge reuse.

---

[3]For example, given the chemical equation of Fischer esterification of acetic acid and ethanol: $CH_3COOH + C_2H_5OH \rightarrow CH_3COOC_2H_5 + H_2O$, MolR assumes the $e_{CH_3COOH} + e_{C_2H_5OH} = e_{CH_3COOC_2H_5} + e_{H_2O}$ also holds, where $e(\cdot)$ represents molecule embedding function.

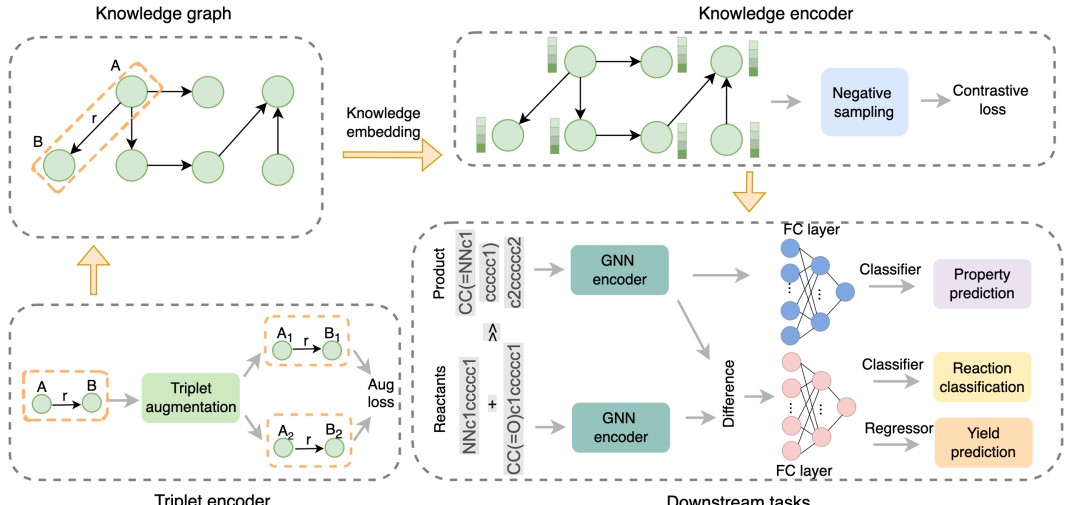

Figure 1: Overview of ReaKE. Our model joint learns molecular embeddings at the triplet-level and knowledge embeddings at the graph-level. The pre-trained model is evaluated on three representative tasks, including reaction classification, yield prediction and molecular property prediction.

## 2.1 DEFINITION OF CHEMICAL SYNTHESIS KNOWLEDGE GRAPH

Chemically speaking, the flow between reactants and products needs to be triggered by environmental conditions such as temperature and pressure, and the reaction conditions are finally reflected on the reaction site which can be represented as the reaction template[4]. Therefore, the introduction of template information can help avoid the ***abnormal energy flow*** problem. In this sense, we reformulate the molecular SSL task as a multi-scale knowledge graph embedding (KGE) task Wang et al. (2017).

We first introduce a Chemical Synthesis Knowledge Graph (CSKG) that is constructed by chemical reaction data. We define the reactant set $R = \{r_1, r_2, ...\}$ as head entities set, the product set $P = \{p_1, p_2, ...\}$ as tail entities set, and the reaction template set $T = \{t_1, t_2, ...\}$ as relations set. Then, we will have a triplet set $B = \{b_1, b_2, \cdots\}$, where $b_i = (r_i, t_i, p_i)$. Triplets in $B$ form the CSKG. For example, as shown in Fig. 2(a), if we have a reaction $C_8H_8O + C_6H_8N_2 \rightarrow C_{14}H_{14}N_2$, we can extract its template $R_1 - CO - R_2 + R_3 - NH_2 \rightarrow R_1R_2 - CN - R_3$ as a relation and build triplets $(C_8H_8O, template, C_{14}H_{14}N_2)$, $(C_6H_8N_2, template, C_{14}H_{14}N_2)$. More detailed information about CSKG is in Appendix B. To integrate the multi-scale representations of molecules and knowledge graphs, we utilize joint contrastive learning to fuse the heterogeneous information.

## 2.2 JOINT CONTRASTIVE LEARNING FOR TRIPLETS AND THE KNOWLEDGE GRAPH

In this section, we introduce our joint learning of triplet-level molecular representation and graph-level knowledge representation. The triplet-level learning is on the individual triplet and the graph-level learning is on a batch of triplets. The main purpose of triplet-level learning is to construct a relatively dense and smooth embedding space for triplets while the graph-level learning aims at improving the model's ability of distinguishing noise samples and making the training more effective. Next, we will explain in detail.

### 2.2.1 MOLECULAR REPRESENTATION LEARNING AT THE TRIPLET-LEVEL

**Molecular Encoder with Graph Neural Network.** Let $\mathcal{G} = (\mathcal{V}, \mathcal{E})$ denote a molecular graph with atoms $\mathcal{V} = \{v_1, v_2, \cdots\}$ and bonds $\mathcal{E} = \{d_1, d_2, \cdots\}$. Atom attributes are set as element type, total

---

[4]The reaction template expresses the reaction mechanism and shows the structural changes before and after the reaction. Its change characteristics can be the standard of reaction classification.

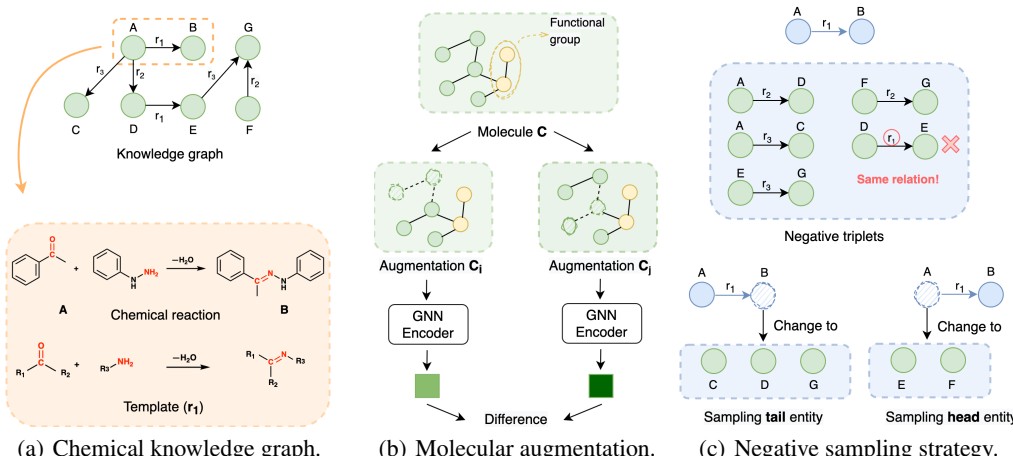

(a) Chemical knowledge graph.  (b) Molecular augmentation.  (c) Negative sampling strategy.

Figure 2: Key points for ReaKE. (a) The construction of chemical synthesis knowledge graph, atoms with red color are reaction center. (b) The reaction-aware negative sampling strategy in a batch. (c) The self-supervised learning with functional group-based augmentation for molecules.

degree, atom is in ring or not, the number of connected hydrogen atoms, atom is aromatic, valence, radical electrons and so on. These atom properties are represented as a one-hot vector and we obtain a total of 69-dimensional atomic features. GNNs utilize the graph connectivity as well as node features to learn representations of atoms and further the entire molecule. Generally, GNNs follow a message passing paradigm, in which each atom representation is iteratively updated by aggregating the representations of its neighbor atoms. At the $l^{th}$ layer, GNN updates the atom embedding $h_i$ of atom $v_i$ as:

$$h_{v_i}^{(l)} = Update^{(l)}(h_{v_i}^{(l-1)}, Aggregate^{(l)}(\{h_{v_j}^{(l-1)}|v_j \in \mathcal{N}(v_i) \cup \{v_i\}\})), \quad (1)$$

where $\mathcal{N}(v_i)$ is the neighbor set of atom $v_i$. $Aggregate(\cdot)$ is an aggregation function depending on the architecture of GNN, more details are in Appendix A. After $L$ layers, the final representation of molecule $\mathcal{G}$ is defined as follows:

$$e_{\mathcal{G}} = Readout(\{h_{v_i}^{(L)}|v_i \in \mathcal{V}\}), \quad (2)$$

where $Readout$ function collects the representation of all atoms and obtains the final molecule representation by pooling operation.

**Triplet-level Contrastive Learning.** As shown in the triplet encoder of Fig. 1.For an individual triplet, we generate more possible triplets to make the chemical embedding space denser and smoother. Thus, small shifts in the embeddings of reactants and products would not affect the correctness of triplets and improve the generalized ability on unseen cases. Specially, we obtain an augmented version of a triplet by keeping the template (relation) and generating a new reactant (head entity) and a new product (tail entity). We keep the reaction center functional group and drop atoms (randomly or crop from edge to center) outside the reaction center with a ratio $\beta$. An illustrative example of molecular augmentation is shown in Fig. 2(b).

For a triplet $(r, t, p)$, the augmentation is applied twice on reactant $r$ and product $p$ separately, resulting in 2 new triplets $(r_{(1)}, t, p_{(1)})$ and $(r_{(2)}, t, p_{(2)})$. After molecular embedding and template embedding (section 2.2.2), we get $(e_{r(1)}, e_t, e_{p(1)})$ and $(e_{r(2)}, e_t, e_{p(2)})$, as we assume $e_r + e_t = e_p$, the augmentation loss is defined as follows:

$$L_{Aug} = |s(e_{p(1)}, e_{p(2)}) - s(e_{r(1)}, e_{r(2)})|, \quad (3)$$

where $s$ is the distance function, $s(e_{r(1)}, e_{r(2)}) = \|e_{r(1)} - e_{r(2)}\|_2$ denotes the $l_2$ norm between two embeddings.

### 2.2.2 KNOWLEDGE REPRESENTATION LEARNING AT THE GRAPH-LEVEL

In addition to the training at the triplet level, we further leverage a contrastive knowledge embedding at the graph level to improve the model's ability of distinguishing noise samples and make the training more effective.

**Template Encoder.** To describe the inference relation from reactant molecule to product molecule, we utilize reaction templates to represent the changes that occur in the chemical process. An illustrative template example has been shown in Fig. 2(a). In particular, we leverage the toolkit RDChiral(Coley et al., 2019) to extract templates with radius 1. Given a reaction $r_1 + r_2 + \cdots + r_k \to p_1 + p_2 + \cdots + p_q$, $\{r_1, r_2, \cdots, r_k\}$ is the reactant set and $\{p_1, p_2, \cdots, p_q\}$ is the product set, we defined the template $t_{r_1} + t_{r_2} + \cdots + t_{r_k} \to t_{p_1} + t_{p_2} + \cdots + t_{p_q}$ and a GNN encoder $g(\cdot)$, the relation embedding is defined as follows:

$$e_t = \sum_{i=1}^{k} g(t_{r_i}) - \sum_{j=1}^{q} g(t_{p_j}), \tag{4}$$

where $k$ denotes the number of reactants, $n$ is the number of products. Note that the GNN encoder $g(\cdot)$ we used here is different from the molecule encoder.

**Graph-Level Contrastive Learning.** In addition, we also propose contrastive strategies for knowledge graph embedding. In particular, we focus on the design of negative sampling Zhang et al. (2019). Negative sampling is a crucial part of the KGE training process, the generation of *hard* negative samples can greatly improve the efficiency and quality of training. However, previous approaches employ indiscriminate negative sampling in a batch, which can easily generate uninformative or wrong negatives. Thus, we propose a reaction-aware negative sampling to avoid same class negatives, which is shown in Fig. 2(c). The negative sampling strategy is detailed as follows:

Denote a mini-batch $B = \{b_1, b_2, \cdots, b_n\}$ of size $n$. For a triplet $b_i = (r_i, t_i, p_i) \in B$, we sample the head entity $r_i'$ or tail entity $p_i'$ in the rest triplets to construct negative triplets set $B_i'$. Head entities and tail entities in triples whose relation are $t_i$ are excluded, that is we do not sample molecules of the same class as $r_i$ or $p_i$. The set $B'$ used to construct negative samples can be formulated as:

$$B' = \{b_k \in B | b_k = (r_k, t_k, p_k), t_k \neq t_i\} \tag{5}$$

The entity set in $B'$ is denoted as $\{R', P'\}$ where the reactant entity set and the product entity set are represented as $R'$ and $P'$, respectively. The negative triplets set $B_i'$ can then be defined as follows:

$$B_i' = \{(r_i', t_i, p_i) | r_i' \in R'\} \cup \{(r_i, t_i, p_i') | p_i' \in P'\} \tag{6}$$

After filtering out same class negatives, we utilize TransE(Bordes et al., 2013) as the basic training objective because it is effective and simple in capturing asymmetry, inversion, and composition relations. Thus, the overall KG loss is defined as follows:

$$L_{KG} = \frac{1}{n} \sum_{i=1}^{n} d_{t_i}(e_{r_i}, e_{p_i}) + \frac{1}{m} \sum_{j=1}^{m} \sigma(\gamma - d_{t_i}(e_{r_j}', e_{p_j}')) \tag{7}$$

where $m$ is the number of negative samples, it is up to $2(n-1)$. $\sigma$ means the sigmoid function, which helps avoid learning easy triplets. $(r_j', t_i, p_j')$ is the $j^{th}$ negative triplet in $B_i'$. $\gamma$ is the margin hyperparameter and the distance function $d_t(e_r, e_p) = \|(e_r + e_t) - e_p\|_2$.

$$d_t(e_r, e_p) = \|(e_r + e_t) - e_p\|_2 \tag{8}$$

### 2.3 OPTIMIZATION OBJECTIVES

Overall, the model is trained jointly with a weighted sum of knowledge embedding loss and molecule augmentation loss mentioned above, where $\lambda$ stands for the trade-off parameter. It is described as follows:

$$L = L_{KG} + \lambda L_{Aug} \tag{9}$$

## 3 EXPERIMENTS

In this section, we conduct various experiments to demonstrate the generality of our model. We first use the embedding-based chemical reaction prediction task to validate whether the framework effectively solves the ***abnormal energy flow*** and the ***ambiguous embedding*** issues. Then, we introduce two reaction-related downstream tasks: reaction classification and yield prediction, to investigate whether our model could capture the changes between the before and after of a reaction (template information) without prior information. Finally, we explore the generalization ability of ReaKE through molecular property prediction tasks. The details of the datasets we used in the pretext and downstream tasks are shown in Appendix B.

### 3.1 CHEMICAL REACTION PREDICTION

**Baselines.** Following the evaluation protocol of MolR(Wang et al., 2021), we compare our models with several state-of-the-art molecular representation methods, including Mole2vec(Jaeger et al., 2018), MolBERT(Li & Jiang, 2021), and MolR. In the baselines, reactants and products are embedded by a molecule encoder, and the dot product of two embeddings is used for ranking product candidates. The pre-training setup can be found in Appendix C.

**Evaluation Protocol.** The reaction prediction task aims to predict the real product's ranking in all candidates of given reactants SMILES. In the test set, considering the direct use of templates or reaction types will lead to the problem of product data leakage, which is not conducive to our product ranking task, we split the template into reactant templates $\{t_{r_1}, t_{r_2}, \cdots, t_{r_k}\}$ and product template $\{t_{p_1}, t_{p_2}, \cdots, t_{p_q}\}$. All products and its $t_p$ in the test set are treated as candidates. For reactants in a reaction, we calculate the embedding distance between $(e_r + \sum_{i=1}^{k} g(t_{r_i}))$ and all product candidates' $(e_p + \sum_{j=1}^{q} g(t_{p_j}))$, and rank the candidates by distance. Then, the true product's ranking is used to calculate mean reciprocal rank (MRR), mean rank (MR), and top-k hit ratio (Hit@k) which are standard evaluation metrics in KG models. Higher MRR, higher Hit@k and lower MR indicate that model achieves a better performance. Note that the results from Mol2vec to MolR are from (Wang et al., 2021).

Table 1: Results of product ranking prediction.

| Methods | MRR | MR | Hit@1 | Hit@3 | Hit@5 | Hit@10 |
|---|---|---|---|---|---|---|
| Mol2vec | 0.681 | 483.7 | 0.614 | 0.725 | 0.759 | 0.798 |
| Mol2vec-FT1 | $0.688 \pm 0.000$ | $417.6 \pm 0.1$ | $0.620 \pm 0.000$ | $0.734 \pm 0.000$ | $0.767 \pm 0.000$ | $0.806 \pm 0.000$ |
| MolBert | 0.708 | 460.7 | 0.623 | 0.768 | 0.811 | 0.858 |
| MolBert-FT1 | $0.731 \pm 0.000$ | $457.9 \pm 0.0$ | $0.649 \pm 0.000$ | $0.790 \pm 0.000$ | $0.831 \pm 0.000$ | $0.873 \pm 0.000$ |
| MolBERT-FT2 | $0.776 \pm 0.000$ | $459.6 \pm 0.2$ | $0.708 \pm 0.000$ | $0.827 \pm 0.000$ | $0.859 \pm 0.000$ | $0.891 \pm 0.000$ |
| MolR | $0.918 \pm 0.000$ | $27.4 \pm 0.4$ | $0.882 \pm 0.000$ | $0.949 \pm 0.001$ | $0.960 \pm 0.001$ | $0.970 \pm 0.000$ |
| ReaKE-SAGE | $0.953 \pm 0.001$ | $4.1 \pm 0.2$ | $0.930 \pm 0.001$ | $0.973 \pm 0.001$ | $0.980 \pm 0.000$ | $0.987 \pm 0.000$ |
| ReaKE-GAT | $0.965 \pm 0.001$ | $6.6 \pm 0.4$ | $0.946 \pm 0.001$ | $0.982 \pm 0.001$ | $0.986 \pm 0.001$ | $0.990 \pm 0.000$ |
| ReaKE-GCN | $0.966 \pm 0.000$ | $4.5 \pm 0.1$ | $0.948 \pm 0.000$ | $\mathbf{0.983 \pm 0.000}$ | $0.987 \pm 0.001$ | $0.991 \pm 0.000$ |
| ReaKE-TAG | $\mathbf{0.967 \pm 0.000}$ | $\mathbf{2.9 \pm 0.0}$ | $\mathbf{0.950 \pm 0.000}$ | $0.982 \pm 0.000$ | $\mathbf{0.987 \pm 0.000}$ | $\mathbf{0.992 \pm 0.000}$ |

**Results.** As shown in Tab. 1, our ReaKE gains about 4.9% MRR and 6.8% Hit@1 performance enhancement against the baseline model MolR and outperforms all Bert-based models. In addition, ReaKE outperforms the baseline regardless of which GNN is combined. Appendix F shows the dice similarity between the predicted product and ground-truth product and cases where our method predicted correctly but other methods did not. Taken together, these results indicate that our method can add triggering conditions between reactants and products, avoid the situation of predicting products as reactants and can learn the changes between reactants and products, which also confirms the effectiveness of introducing templates.

### 3.2 REACTION CLASSIFICATION

**Baselines and Evaluation Protocol.** The goal of reaction classification task is to predict the category of the given reaction. We consider several molecule fingerprints and reaction fingerprints as baselines, including AP3(Carhart et al., 1985), DRFP(Schneider et al., 2015), RXNFP(Schwaller et al., 2021a) and RxnRep(Wen et al., 2022). AP3 is an atom-pairs molecular fingerprint method

with a maximum path length of three. DRFP creates a hash binary fingerprint based on the symmetric difference between substructures.

To evaluate the effectiveness of the learned representations, we use our pre-trained model as a feature extractor and obtain the final reaction representations by calculating the difference between reactants and products. Then, we train MLP for reaction classification. Following the few-shot setting of RxnRep(Wen et al., 2022), instead of using the entire training set, we sample 4, 8, 16, 32, 64, 128 reactions per class to simulate the situation of a small dataset. Every experiment is repeated five times with the resampling of training data. In addition, we visualize the initial embedding space directly encoded by the pre-trained model. We demonstrate the distribution of chemical reaction embeddings with different classes and the distribution of reactants and products within a type of reaction. Note that the results from AP3 to RxnRep are from(Wen et al., 2022). For MolR, we use the provided model.[5]

Table 2: Classification $F_1$ score on the Schneider dataset.

| Methods | 4 reactions per class | 8 reactions per class | 16 reactions per class | 32 reactions per class | 64 reactions per class | 128 reactions per class |
|---|---|---|---|---|---|---|
| AP3 | $0.518 \pm 0.004$ | $0.620 \pm 0.004$ | $0.703 \pm 0.006$ | $0.761 \pm 0.002$ | $0.799 \pm 0.004$ | $0.828 \pm 0.004$ |
| DRFP | $0.100 \pm 0.005$ | $0.129 \pm 0.004$ | $0.199 \pm 0.008$ | $0.266 \pm 0.007$ | $0.338 \pm 0.006$ | $0.398 \pm 0.002$ |
| RXNFP | $0.322 \pm 0.012$ | $0.394 \pm 0.013$ | $0.471 \pm 0.010$ | $0.531 \pm 0.006$ | $0.575 \pm 0.005$ | $0.618 \pm 0.004$ |
| RxnRep | $0.441 \pm 0.010$ | $0.634 \pm 0.003$ | $0.767 \pm 0.003$ | $0.831 \pm 0.002$ | $0.875 \pm 0.003$ | $0.900 \pm 0.002$ |
| MolR | $0.629 \pm 0.007$ | $0.722 \pm 0.009$ | $0.803 \pm 0.006$ | $0.862 \pm 0.006$ | $0.901 \pm 0.004$ | $0.900 \pm 0.027$ |
| ReaKE-SAGE | $0.796 \pm 0.011$ | $0.860 \pm 0.010$ | $0.892 \pm 0.004$ | $0.908 \pm 0.003$ | $0.918 \pm 0.003$ | $0.933 \pm 0.002$ |
| ReaKE-GAT | $0.778 \pm 0.012$ | $0.844 \pm 0.004$ | $0.876 \pm 0.005$ | $0.897 \pm 0.005$ | $0.917 \pm 0.003$ | $0.923 \pm 0.007$ |
| ReaKE-GCN | $0.765 \pm 0.008$ | $0.842 \pm 0.008$ | $0.877 \pm 0.009$ | $0.898 \pm 0.003$ | $0.918 \pm 0.004$ | $0.923 \pm 0.016$ |
| ReaKE-TAG | $\mathbf{0.821 \pm 0.011}$ | $\mathbf{0.882 \pm 0.003}$ | $\mathbf{0.901 \pm 0.005}$ | $\mathbf{0.916 \pm 0.004}$ | $\mathbf{0.928 \pm 0.003}$ | $\mathbf{0.935 \pm 0.004}$ |

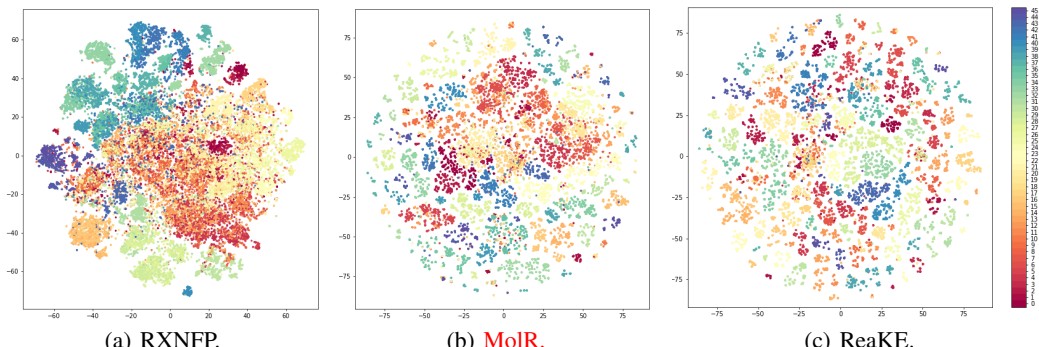

(a) RXNFP.        (b) MolR.        (c) ReaKE.

Figure 3: t-SNE visualization of RXNFP, MolR and ReaKE on schneider dataset. (a), (b)and (c)show the fingerprint distribution of the entire schneider dataset, with 46 colors representing 46 categories of reactions.

**Results.** The results under transfer feature extraction setting are illustrated in Tab. 2, ReaKE achieves 19.2%, 16%, 9.8%, 5.4%, 2.7%, 3.5% $F_1$ score gain over the state-of-art model with 4, 8, 16, 32, 64, 128 reactions per class as the training set. It demonstrates our learned embeddings can transfer well to downstream tasks with a small training set. And our model could capture the changes between the before and after of a reaction (template information) without prior information.

t-SNE visualizations of RXNFP, MolR and our method on the Schneider dataset are shown in Fig. 6. We perform two kinds of visualizations: One is the visualization on the reaction fingerprint level as shown in Fig. 3(a), Fig. 3(b) and Fig. 3(c), that is obtain the embeddings of all reactions from pre-trained model, and then conduct t-SNE visualization to see the distribution of the initial embeddings in the embedding space. As shown in Fig. 3(c), there is a clear boundary between each category and almost no overlapping parts compared with RXNFP and MolR. Furthermore, the reactions of the same category are clustered in multiple clusters, which demonstrates that our reaction representations can also capture discriminative properties about reaction templates. The other is the visualization on the molecular level which can be found in Appendix D.

---

[5]https://github.com/hwwang55/MolR/tree/master/saved

## 3.3 YIELD PREDICTION

**Baselines and Evaluation Protocol.** Yield prediction aims at predicting the yield of a given reaction. For comparison, we choose the DFT-based method, DRFP(Schneider et al., 2015), Yield-BERT and its augmented version Yield-BERT (aug.) (Schwaller et al., 2020) as baselines. Yield-BERT is an extension of the learned RXNFP fingerprint with a regression layer.

Table 3: $R^2$ of yield prediction on Buchwald Hartwig reactions.

|  | Test 1 | Test 2 | Test 3 | Test 4 | Avg. 1-4 |
|---|---|---|---|---|---|
| DFT | 0.80 | 0.77 | 0.64 | 0.54 | 0.69 |
| DRFP | $0.81 \pm 0.010$ | $0.83 \pm 0.003$ | $0.71 \pm 0.001$ | $0.49 \pm 0.004$ | $0.71 \pm 0.160$ |
| Yield-BERT | $0.84 \pm 0.010$ | $0.84 \pm 0.030$ | $\mathbf{0.75 \pm 0.040}$ | $0.49 \pm 0.050$ | 0.73 |
| Yield-BERT(aug.) | $0.80 \pm 0.010$ | $0.88 \pm 0.020$ | $0.56 \pm 0.080$ | $0.43 \pm 0.040$ | $0.58 \pm 0.330$ |
| MolR | $0.68 \pm 0.003$ | $0.84 \pm 0.002$ | $0.61 \pm 0.006$ | $0.51 \pm 0.003$ | $0.66 \pm 0.120$ |
| ReaKE | $\mathbf{0.87 \pm 0.002}$ | $\mathbf{0.89 \pm 0.002}$ | $0.67 \pm 0.004$ | $\mathbf{0.57 \pm 0.005}$ | $\mathbf{0.75 \pm 0.140}$ |

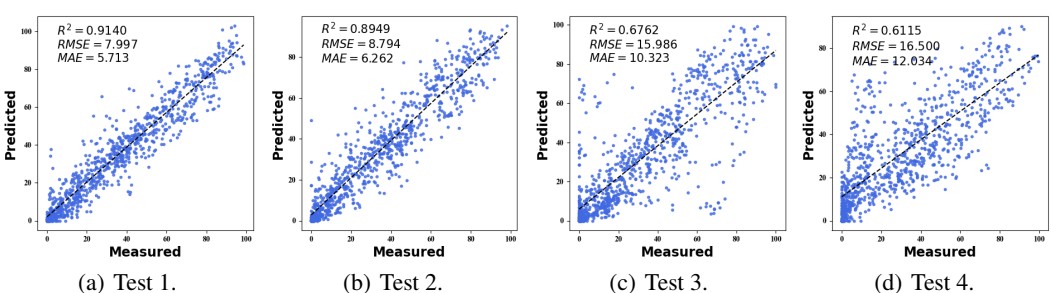

(a) Test 1.     (b) Test 2.     (c) Test 3.     (d) Test 4.

Figure 4: Fine-tune performance $R^2$ scores (%) on Test1-4 of ReaKE.

We obtain the reaction representations from our pre-trained model, and then feed it into xgboost(Chen & Guestrin, 2016). We leveraged the four tests in DRFP(Schneider et al., 2015) to show the results under splits based on isoxazole additives with 5 times run. We report the mean and standard deviation of $R^2$ score. Note that the results from DFT to Yield-BERT(aug.) are from (Schneider et al., 2015).

**Result.** We apply feature extraction setting and fine-tune setting on the yield prediction task. The results under two settings are shown in Tab. 3 and Fig. 4 separately. For the feature extraction, our method gains about 2% average $R^2$ value enhancement on Test1-4. For the fine-tuning setting, we achieve a 4% average $R^2$ improvement on Test1-4. In addition, in Test1 and Test4, our fine-tune model gains about 7% $R^2$ improvement. In conclusion, our method can capture subtle changes in the reaction and improve the performance of prediction yield prediction, which also confirms that pretext template information is beneficial for downstream tasks.

## 3.4 MOLECULAR PROPERTY PREDICTION

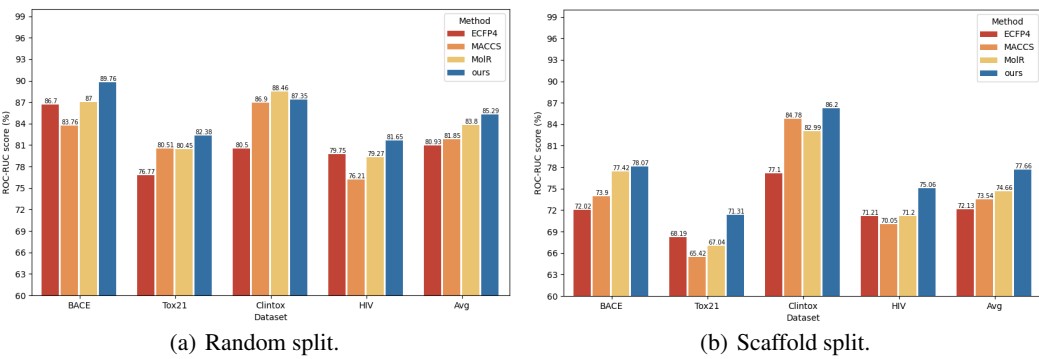

(a) Random split.        (b) Scaffold split.

Figure 5: Fine-tune performance ROC-AUC scores (%) on property prediction benchmarks.

**Baselines and Evaluation Protocol.** The goal of molecular property prediction is to predict labels of given molecules. To further confirm our pre-trained model's potential when addressing tasks not related to chemical synthesis, we apply ReaKE to molecular property predictions. We select the rule-based methods ECFP4(Rogers & Hahn, 2010), MACCCS(Heikamp & Bajorath, 2011), and the self-supervised learning method MolR as baselines. Due to the large gap between the pretext dataset and property datasets, we verify our method by adding an MLP layer under the fine-tuning setting. We demonstrate the average ROC-AUC score of the datasets under random splits and scaffold splits, we apply 5-fold cross-validation to random splits and run experiments 5 times with different random seeds under scaffold splits.

**Results.** The evaluation results under fine-tune setting are illustrated in Fig. 5. On average, we achieve a 1.5% and 3.0% gain on the random splits and scaffold splits. It suggests the representation can also transfer well to molecular-related tasks.

# 4 RELATED WORK

**Sequence-based methods.** One line of works to represent molecules/reactions are BERT-based(Devlin et al., 2018) or Transformer-based(Vaswani et al., 2017) models, such as the SMILES-BERT(Wang et al., 2019), Mol-BERT(Li & Jiang, 2021), Mol-Transformer(Schwaller et al., 2019), RXNFP(Schwaller et al., 2021a), K-BERT(Wu et al., 2022) and KV-PLM(Zeng et al., 2022). These methods treated SMILES sequences of molecules/reactions as text and generated efficient molecular/reaction embeddings by designing molecule-made-to-measure BERT or Transformer models. These methods only use the sequence information of SMILES, however, molecular biological activity largely depends on its structure.

**Structure-based methods.** The other line of studies focus on topological information of molecules. Some traditional methods are topology-based hashed fingerprints, such as AP3 (Carhart et al., 1985), ECFP4(Rogers & Hahn, 2010) and DRFP(Schneider et al., 2015), they capture the extended-connectivity and leverage the linear path or radial substructure of molecules/reactions. And recent studies mostly leverage GNN-based methods to model the 2D structure of molecules. Among them, self-superveised Learning (SSL)methods are actively proposed to maximize the use of unlabeled data, such as GraphCL(You et al., 2020), MICRO-Graph(Zhang et al., 2020), GraphLoG(Xu et al., 2021), MolCLR(Wang et al., 2022) and KPGT(Li et al., 2022). The above methods leverage contrastive learning on molecules and focus on extracting subgraph patterns for more comprehensive molecular representations.

**Knowledge-based SSL Methods.** However, recent studies have pointed out that pre-trained GNNs with node prediction, context prediction, and motif prediction pretext tasks gives limited improvements and often lead to negative transfer on downstream tasks (Hu et al., 2020; Stärk et al., 2021; Sun, 2022). Thus, It is necessary to provide domain knowledge and design complex pretext task since this can force model learn more useful information. Recent studies inject extra chemical reaction knowledge into SSL training to empower the learned embeddings. For example, RxnRep (Wen et al., 2022) leverages the chemical reaction data and makes the two augmented representations of a reaction similar to each other but distinct from different reactions. MolR(Wang et al., 2021) preserves the equivalence of molecules with respect to chemical reactions in the embedding space.

# 5 CONCLUSION

In this work, we propose a simple yet effective chemical synthesis knowledge graph to tackle the challenges in modeling reaction data. We introduce the changes in reaction sites as the trigger conditions of flow between molecules and build explicit connections between molecules through reaction template information. Besides, we joint learning the triplet-level molecular representation and the graph-level knowledge representation. Comprehensive experiments over multiple benchmark downstream tasks consistently demonstrate the effectiveness of our method.

Further directions to explore can be described as the following points: First, we can take into account the 3D structure of molecules and reactions. Second, it is interesting to consider external factors related to the reaction (i.e., temperature, reaction conditions, etc.).

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

# A    DETAILS OF GNN ARCHITECTURES

We use SMILES string of molecules as input and then change SMILES into Mol through "rd-kit.Chem"[6], finally access atoms and edges in Mol and build graphs. The molecular graphs are the input of GNN encoders. Following MolR(Wang et al., 2021), we apply 4 kinds of GNN architecture based on Deep Graph Library (DGL)[7]. The above GNNs all follow a message-passing framework, i.e., at each iteration, every node aggregates information from its local neighbor, thus we illustrate the detailed message passing pattern of the GNNs:

**Graph Convolutional Network (GCN)**(Kipf & Welling, 2016). In GCN, the features of neighbors are summed directly, that is, we treat the neighbors of the current node equally.The feature matrix of the $l^{th}$ layer are as follows:

$$H^{(l)} = \sigma(\widetilde{D}^{-\frac{1}{2}}\widetilde{A}\widetilde{D}^{-\frac{1}{2}}H^{(l-1)}W^{(l-1)}), \tag{10}$$

where $W^{(l-1)}$ is the weight matrix at $(l-1)$ layer, $\widetilde{A}$ is the adjacency matrix with added self-loop, $\widetilde{D}_{kk} = \sum_j \widetilde{A}_{kj}$, and $\sigma(\cdot)$ means the activation function.

**Graph Attention Network (GAT)**(Velickovic et al., 2017). In GAT, it aggregates neighbor nodes through the attention mechanism, and realizes the adaptive allocation of different neighbor weights. The feature of node $i$ at the $l^{th}$ layer is:

$$h_i^{(l)} = \sigma(\sum_{j \in \mathcal{N}(i)} \alpha_{ij}W^{(l-1)}h_j^{(l-1)}), \tag{11}$$

where $\alpha_{ij}$ represents the attention score between node $i$ and its neighbor $j$, $\alpha_{ij} = softmax_j(e_{ij})$ and $e_{ij} = a(Wh_i, Wh_j)$, $a(\cdot)$ is a shared attentional mechanism.

**Graph Sample and Aggregate (GraphSage)**(Hamilton et al., 2017). In GraphSage, it learns the embeddings of each node in an inductive mode and updates embedding by sampling and aggregating. It can naturally generalize to unseen nodes.

$$\begin{aligned} h_{\mathcal{N}(i)}^{(l)} &= Aggregate_l(\{h_v^{(l-1)}, \forall v \in \mathcal{N}(i)\}), \\ h_i^{(l)} &= \sigma(W^{(l)} \cdot Concat(h_i^{(l-1)}, h_{\mathcal{N}(i)}^{(l)})), \end{aligned} \tag{12}$$

where $Aggregate(\cdot)$ can be mean aggregator, LSTM aggregator and pooling aggregator.

**Topology Adaptive Graph Convolutional Network (TAGCN)**(Du et al., 2017). It is a variant of GCN, TAGCN uses $K$ graph convolution kernels to extract local features of different sizes, and retains $K$ as a hyperparameter.

$$H^{(l)} = \sigma(\sum_{k=0}^{K}(D^{-\frac{1}{2}}AD^{-\frac{1}{2}})^k H^{(l-1)}\Theta_k^{(l)}), \tag{13}$$

where $K$ stands for the number of local filters, $\Theta_k^{(l)}$ is a weight matrix of the $k$ hop. Totally, TAGCN can extract local features on a set of size-1 up to size-K receptive fields.

# B    DATASETS

**Pre-training Dataset.** The dataset we leverage for constructing the chemical synthesis knowledge graph here is USPTO-479k(Zheng et al., 2019). It contains reactions with up to five reactants and only one product. By removing reactions from which we cannot extract templates and others that contain the same reactants and products, we finally obtain 407,039 training reactions, 29,848 validation reactions and 39,802 testing reactions. We further convert the training reactions into 587,403 triplets with 103,339 reaction templates.

**Downstream Task Datasets.** For the reaction classification task, we use the Schneider dataset (Schneider et al., 2015). It is derived from the Schneider 50k dataset, which is a descendant of the

---

[6]http://rdkit.org/docs/source/rdkit.Chem.html
[7]https://www.dgl.ai/

USPTO dataset of patent reactions. After further cleaning, we obtain 38,800 reactions with 46 reaction types. We split it into a training set with 31,002 reactions, a validation set with 3,896 reactions and a test set with 3,902 reactions. For the yield prediction task, a palladium-catalyzed Buchwald-Hartwig C-N cross-coupling reactions dataset(Ahneman et al., 2018) is utilized to evaluate model performance in our experiment. It includes 3,955 reactions labeled with yield. These reactions are composed of 15 aryl halides, 1 methylaniline, 4 Buchwald ligands, 1 Pd catalyst, 3 bases and 23 additives. In addition, following drfp(Probst et al., 2022), we used four out-of-sample splits based on isoxazole additives and created a 70/10/20 train/valid/test split. For the molecular property prediction task, we use four Open Graph Benchmark (OGB) datasets with their standard scaffold splits and random splits. The datasets are BACE, Tox21, Clintox and HIV dataset(Wu et al., 2018).

## C  PRE-TRAINING SETUP

We consider four GNN models as molecule encoders, they are GCN(Kipf & Welling, 2016), GAT(Velickovic et al., 2017), SAGE(Hamilton et al., 2017), and TAG(Du et al., 2017). In all models, we use a 2-layer GNN with a sum pooling Readout function and project the representation to a 1024-dimensional latent space for both molecule encoder and template encoder. Besides, we optimize our model using Adam optimizer with a learning rate of 5e-5. As a default setting, we use a margin value $\gamma$ of 4.0, a molecule augmentation drop ratio $\beta$ of 0.7, and a trade-off parameter $\lambda$ of 1.0. Furthermore, we train at batch-size 1024 for 30 epochs on a NVIDIA GeForce GTX 1080Ti GPU. Pre-training on 587k triplets of CSKG takes 5.8 hours.

Our pre-training model consists of a Graph Isomorphism Network (GIN) from **?** with 5 layers and 300 hidden dimensions and a residual convolutional neural network (ResNet-34) **?** with 63.5M parameters. We pre-train the model for 100 epochs using a batch size of 1024 on 8 NVIDIA 3090TI GPUs. We use the Adam optimizer with an initial learning rate of 3e-4 and weight decay of 0.02. We take image with resolution of 128×128. The margin $\gamma$ is set to 4. Pre-training on 750k graph-image pairs for MIGA takes 8 hours, far less than 26 hours for ContextPred and 48 hours for GraphCL on 280k molecules of GEOM-Drugs.

## D  VISUALIZATION ON SCHNEIDER DATASET

We further demonstrate the distribution of reactants and products within a type of reaction in Schneider dataset.

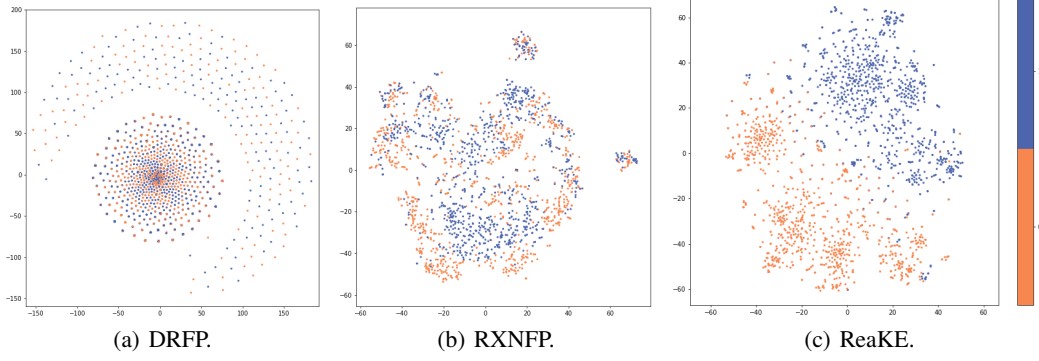

| (a) DRFP. | (b) RXNFP. | (c) ReaKE. |

Figure 6: t-SNE visualization of ReaKE, DRFP, RXNFP on schneider dataset. (a), (b)and (c) demonstrate the distribution of reactant and product embeddings within the reactions of the same class, label **0** stands for reactants and label **1** stands for products.

The visualization on the molecular level is shown in Fig. 6(a), Fig. 6(b) and Fig. 6(c), which is gaining the embeddings of reactants and products in a class of Schneider reactions, and then using t-SNE visualization to explore the distribution. As shown in Fig. 6(c), the reactants and products form clusters separately, which suggests the ***Ambiguous embeddings*** problem is alleviated, and

there exist differences between the embeddings of the reactant and the product. Besides, our model can aggregate molecules with the same functional group.

# E   ABLATION STUDY

We further analyze the contribution of different components under two kinds of augmentation in our ReaKE. The drop atom mode randomly crops atoms outside the reaction center, and the subgraph mode crops atoms from edge to center. The variants are as follows: (1) **w/o neg**: w/o negative sampling; (2) **w/o aug**: w/o molecular augmentation; (3) **w/o temp**: w/o adding reaction templates information.

The ablation results on chemical reaction prediction are reported in Appendix E. In specific, the removal of the template component leads to the most significant performance drop, which is align with our assumption that template information can benefit molecular modeling. Note that molecular augmentation can bring obvious improvements. We show two kinds of augmentations: drop atom mode and drop subgraph mode. The exclusion of both strategies will decrease the results, showing the importance of augmentation. We also try to iterate the estimation w/o negative sampling but observe a performance drop given more iterations.

Table 4: Ablation study results on chemical reaction prediction.

| Method | MRR | MR | Hit@1 | Hit@10 |
|---|---|---|---|---|
| **ReaKE (drop atom)** | **0.967** | **2.8** | **0.950** | **0.992** |
| -w/o neg | 0.925 | 11.6 | 0.900 | 0.967 |
| -w/o aug | 0.953 | 3.7 | 0.930 | 0.987 |
| -w/o temp | 0.917 | 33.9 | 0.880 | 0.970 |
| **ReaKE (subgraph)** | **0.967** | **3.3** | **0.950** | **0.991** |
| -w/o neg | 0.922 | 18.0 | 0.896 | 0.966 |
| -w/o aug | 0.953 | 3.7 | 0.930 | 0.987 |
| -w/o temp | 0.914 | 40.4 | 0.880 | 0.965 |

# F CASE STUDY

To explore whether our method can effectively solve the ***Ambiguous embeddings*** problem in the pretext task, we statistics the distribution of dice FP similarities between predicted products and real products. We except the cases where both our method and MolR predict the true product for demonstrating the differences more explicitly.

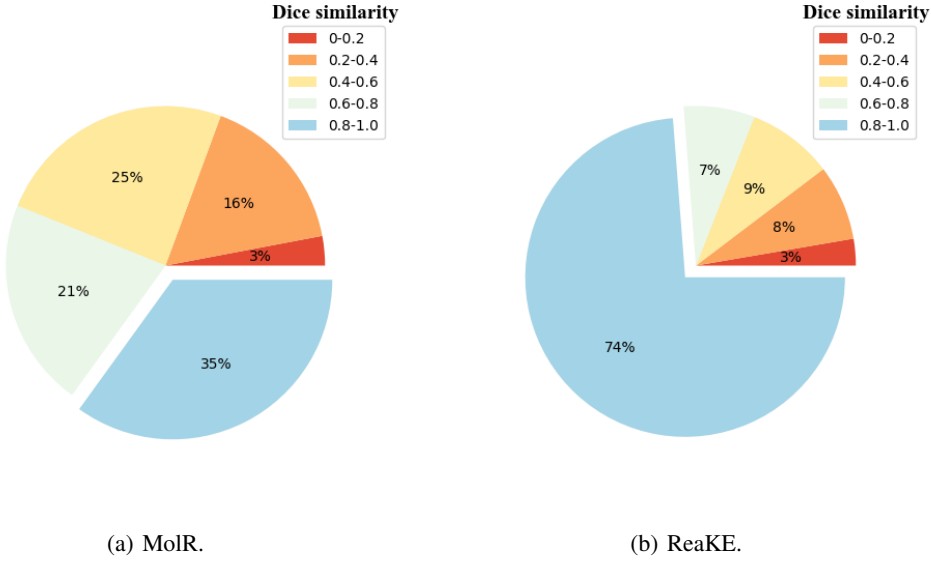

(a) MolR.  (b) ReaKE.

Figure 7: Pie chart of dice FP similarities distribution of predicted and real products.

We also conduct a case study on USPTO-479k and select the first 100 reactions on the test dataset to see if our optimization is effective compared with MolR. The detailed results are demonstrated on Table 5, we remove examples where all methods predict correctly.

Table 5: Case study on the first 100 reactions of USPTO-479k test set.

| Index | Reactant(s) | Ground-truth product | Predicted product by ReaKE | Predicted product by MolR |
|-------|-------------|----------------------|----------------------------|---------------------------|
| 5 | | | same as ground-truth | |
| 10 | | | same as ground-truth | |

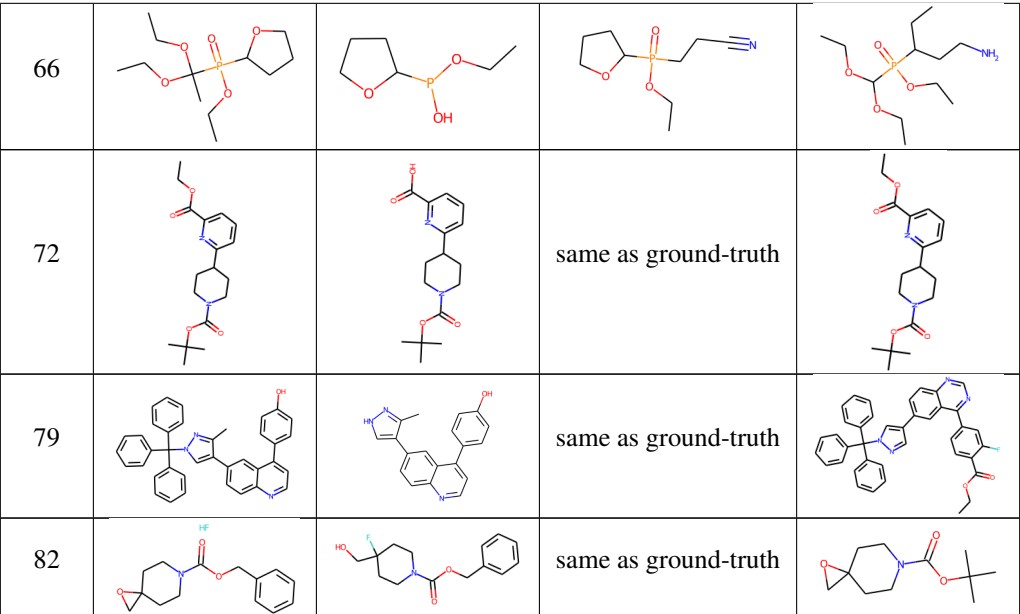

| 66 | | | | |
|---|---|---|---|---|
| 72 | | | same as ground-truth | |
| 79 | | | same as ground-truth | |
| 82 | | | same as ground-truth | |

As shown in Table 5, our method models chemical reaction data in detail and avoids some cases where the predicted product value is consistent with the reactant (such as the No.5 reaction and the No.72 reaction). This phenomenon also confirms that our initial analysis of the problem and optimization measures are effective. Besides, there are reactions (such as the No. of 66 reaction) that cannot be predicted correctly by both methods, which is due to the fact that only the main product is retained in USPTO-479k, and the by-product is omitted. This deficiency leads to inaccurate prediction, but the correct product ranking predicted by our method is still better than previous methods.

