# OpenReview forum: "ReaKE: Contrastive Molecular Representation Learning with Chemical Synthetic Knowledge Graph"
_ICLR.cc/2023/Conference — Submitted to ICLR 2023_

### Official Review · Reviewer_5dFJ · 2022-10-23

**Confidence:** 2
**Correctness:** 4
**Technical Novelty And Significance:** 2
**Empirical Novelty And Significance:** 3
**Recommendation:** 6

**Clarity, Quality, Novelty And Reproducibility:**

The paper is clearly written and the proposed ideas seem to be novel.  However, I think details about the experimental setup are not very well documented in the appendix, publishing the code in the future could be helpful for reproducibility.

**Strength And Weaknesses:**

Strength:
They consider the issues such as abnormal energy flow and ambiguous embeddings which were neglected before and proposed valid ways to address them.  The constructive loss defined on the knowledge graph seems to be intuitive and interesting.


weakness:

**Summary Of The Paper:**

This paper proposes a self-supervised representation method that is based on chemical reaction data. The paper points out that existing methods have disadvantages such as abnormal energy flow,  ambiguous embeddings, and sparse embedding space. To address these disadvantages of the existing reaction-based representation learning, they propose to use a chemical synthetic knowledge graph that connects molecules using reaction templates and individual graphs that represent the 2D structure of the molecules.  They claim that the intorduction of the reaction knowleadge graph can elivate the energy flow and the ambiguous embedding problems. Besides, they propose to use of a functional group-augmented SSL method for reaction triplet representation learning which they believe will solve the sparse embedding space problem.

**Summary Of The Review:**

The paper overall is written clearly, but I have some minor comments or questions as following

1. How does the contrastive knowledge embedding
at the graph, level make the triplet embeddings more evenly distributed on the embedding space? also, there is a typo here in the paper knoledge graph instead of the knowledge graph.



2. I am still a bit confused about the intuition behind this: "The main purpose of triplet-level learning is to construct a
smooth latent embedding space for reactants and products while graph-level learning aims to make
triplet embeddings distributed on the embedding space more evenly"

3. In the section titled "Triplet-level Contrastive Learning",  the loss seems to be simply on the augmentations of the reactant and augmentations of the products, not sure why it is called triplet-level contrastive learning, this title is a bit confusing. I was wondering without any template or triplet, just doing the same augmentation on each of the molecules and applying similar loss would have the same effect. Also in the appendix, the latent of this model seems to separate the reactant and product very well, how did this happen? also, do we really want to separate the reactant and product? because a product in this reaction could be a reactant in another reaction I think.

---

> ### Author Response · Authors · 2022-11-17
> **About clearer purpose of the components in our framework.**
>
> First, we appreciate the reviewer for carefully reading our manuscript and proposing insightful comments.
>
> ## **Reproducibility**:
>
> We have uploaded our code at the supplementary material.
>
> ---
>
> ## **Responses to the questions:**
>
> **Q1: How does the contrastive knowledge embedding at the graph, level make the triplet embeddings more evenly distributed on the embedding space? also, there is a typo here in the paper knoledge graph instead of the knowledge graph.**
>
> **Response**:  Maybe we didn't describe this in enough detail in our previous article. We re-describe the role of **"contrastive knowledge embedding"**. In the Graph-level contrastive learning, we keep the gamma distance of right triplets and wrong triplets through contrastive training (the keep gamma distance is what we want to express about "evenly distributed" which will improve the model's ability of **distinguishing noise samples and making the training more effective**). At the same time, we exclude the triplets that have the same template (relation) of right triplet by reaction-aware negative sampling to avoid **same class negatives**.
>
> ---
>
> **Q2: I am still a bit confused about the intuition behind this: "The main purpose of triplet-level learning is to construct a smooth latent embedding space for reactants and products while graph-level learning aims to make triplet embeddings distributed on the embedding space more evenly"**
>
> **Response**:  Thanks for your valuable comments. Triplet-level learning is for **a single triplet**, which obtains a dense space by generating some new triplets. Graph-level learning is for **a set of triplets** in a batch, we use reaction-aware negative sampling to make the learning on the KG more effective.
>
> Regarding "triplet-level learning", we are trying to **construct a smooth space**, i.e., if given an A->B reaction, in the embedding space there will be e(A) + e(template) = e(B), e (·) represents the embedding function. Suppose there is a small **perturbation **sigma (say removing an atom outside the reaction center on both A and B, we get A' and B'), due to the sparsity of the chemical space, it may cause a large offset, that is e(A') + e(template)! =e(B'). To avoid this situation, we augment the reactants and products of each triplet to generate new possible triplets, then we obtain a relatively dense embedding space that can extend to unknown cases and make e(A') + e(template) = e(B') holds. A detailed description of "graph-level learning" see Q1.
>
> ---
>
> **Q3: In the section titled "Triplet-level Contrastive Learning", the loss seems to be simply on the augmentations of the reactant and augmentations of the products, not sure why it is called triplet-level contrastive learning, this title is a bit confusing. I was wondering without any template or triplet, just doing the same augmentation on each of the molecules and applying similar loss would have the same effect. Also in the appendix, the latent of this model seems to separate the reactant and product very well, how did this happen? also, do we really want to separate the reactant and product? because a product in this reaction could be a reactant in another reaction I think.**
>
> **Response**:  We hope our rebuttal can clarify your concerns.
>
> (1) The reason for the name “Triplet-level Contrastive Learning” is that we **keep the template (relation)**, and then augment the parts except the reaction center of reactant and product to generate a new triplet, so it is called triplet-level learning. i.e., given e(A)+e(template)=e(B), we can generate A1, B1 and A2, B2 by augmenting A and B, we want e(A1)+e(template)=e( B1) and e(A2)+e(template)=e(B2), so the optimization objective can be obtained as e(B1)-e(A1)=e(B2)-e(A2), and the final optimization objective is [e(A1)-e(A2)]-[e(B1)-e(B2)] = 0. If the molecule is augmented indiscriminately, it may destroy the important functional group structure of the molecule, resulting in a large change in the properties of the molecule, which will have a negative effect on our training.
>
> (2) About the visualization in the Appendix, we aim at showing that molecules with the same **functional group** tend to **aggregate**. We take the reactants and products in a class of reactions for visualization, that is, if this class of reactions is an esterification reaction, they all follow the template: R1-COOH + R2-OH -> R3-COO. What our method shows is that molecules with the same functional group are in an aggregated state, that is: in the same type of reaction, the reactants tend to be of the same type (all acids), and the products tend to be of the same type (all esters). Of course, this product might be a reactant in another reaction and occurs in different types of reactions, we only show the situation in exactly one class of reactions.

---

### Official Review · Reviewer_EYxB · 2022-10-25

**Confidence:** 3
**Correctness:** 3
**Technical Novelty And Significance:** 3
**Empirical Novelty And Significance:** 2
**Recommendation:** 5

**Clarity, Quality, Novelty And Reproducibility:**

The clarity of the experiments in this paper is not enough. The quality and novelty of the paper are reasonable. The authors do not provide code for reproducibility.

**Strength And Weaknesses:**

Pros:
1. The paper is well-organized and easy to follow.
2. The proposed approach is simple and useful to deal with the previous limitations.

Cons:
1. The experiments are not clearly described and exhibited:
    - It is unclear what the "TAG" model is. In Appendix A, GCN, GAT, SAGE, and TAG all lack their references.
    - From the results shown in Table 1 and Table 2, it is not necessary to include the results of ReaKE-SAGE/GAT/GCN, since ReaKE-TAG consistently performs better than the other three variants.
    - It is unclear whether the results of the baseline models come from their original papers, or from the authors' own experiments. For example, the results of MolR in Table 1 are directly from its original paper [1]. However, the results of MolR in Table 2 and 3 are not found in [1]. The authors do not mention how they get the results.
    - For the visualization in Figure 3 and 6, the baselines being shown are only DRFP and RXNFP, which perform much worse than the proposed method as shown in Table 2. It would be more convincing to see the visualization of MolR, since it is the most competitive baseline.
    - For the molecule property prediction task, the results of ECFP4 and MolR shown in Figure 5 of this paper and the results in MolR's original paper (Table 3 in [1]) are very different. The authors should clarify this mismatch. Besides, it would be more clear to just use a table to list the values in Figure 5 for comparison.
2. The orders of Figure 2b and Figure 2c can be exchanged to match the orders of the corresponding paragraphs.
3. The authors should double-check if the related papers are correctly referenced. For example, MolR has been already accepted by ICLR 2022 rather than just on arxiv.
4. Typo: In section 2.2.2, "knoledge" --> "knowledge"

[1] Wang, Hongwei, Weijiang Li, Xiaomeng Jin, Kyunghyun Cho, Heng Ji, Jiawei Han, and Martin Burke. "Chemical-Reaction-Aware Molecule Representation Learning." In International Conference on Learning Representations. 2021.

**Summary Of The Paper:**

This work proposes a molecular representation learning method based on pre-training with chemical synthetic knowledge graph. The proposed method tackles and solves the limitations in previous works that use chemical reaction knowledge in self-supervised learning.

**Summary Of The Review:**

The motivation of the work is strong, and the proposed approach is simple and practical to deal with the previous limitations. However, the experiments need more clear descriptions and convincing comparisons.

---

> ### Author Response · Authors · 2022-11-17
> **About how we got the result of baselines.**
>
> We appreciate the reviewer for carefully reading our paper, and we are very glad for the positive feedback and constructive comments.
>
> ## **Responses to the Cons:**
>
> **Q1: The experiments are not clearly described and exhibited:**
>
> **Responses to Q1**:
>
> (1) Thanks for your suggestion and we added a detailed description of GNNs in **Appendix A**.
>
> (2) We report the results of several variants to demonstrate the general advantage of the framework on GNNs and the general scalability on downstream tasks, so we retain the results of GNNs in Tables 1 and 2.
>
> (3) We supplement the corresponding description in **"Baselines and Evaluation Protocol"** part of every task, the results in Tables 2 and 3 are obtained by directly using the model provided by MolR[1] to extract features.
>
> (4) Thank you for your professional suggestion, we added the visualization of MolR in **Figure 3(b)**. The smaller overlap and clearer boundaries between different kinds of reactions on t-SNE visualization compared to MolR illustrates the effectiveness of our method.
>
> (5) The reason of unmatch is we use two kinds of splits which we think are more comprehensive. One is the ogb **scaffold split** [2], and the other is **five-fold cross-validation**. And the results in MolR are obtained by 20 times random split.
>
> ---
>
> **Q2: The orders of Figure 2b and Figure 2c can be exchanged to match the orders of the corresponding paragraphs.**
>
> **Response**: Thanks for your help. We have adjusted the order of the images, which resulted in better clarity in our articles.
>
> ---
>
> **Q3: The authors should double-check if the related papers are correctly referenced. For example, MolR has been already accepted by ICLR 2022 rather than just on arxiv.**
>
> **Response**: Since we paid attention to this article long ago, we forgot to update the reference, we have updated the corresponding content in the References section.
>
> ---
>
> **Q4: Typo: In section 2.2.2, "knoledge" --> "knowledge"**
>
> **Response**: Thank you for your reminding, we have revised the typos throughout the paper.
>
> ---
> [1] https://github.com/hwwang55/MolR/tree/master/saved
>
> [2] https://ogb.stanford.edu/docs/graphprop/

---

### Official Review · Reviewer_ToSM · 2022-11-02

**Confidence:** 4
**Correctness:** 3
**Technical Novelty And Significance:** 2
**Empirical Novelty And Significance:** 2
**Recommendation:** 5

**Clarity, Quality, Novelty And Reproducibility:**

Clarity, Quality: The paper is clear.

Novelty: The method consists of several points which appear in existing works (contrastive learning, augmentation and negative sampling). For molecular augmentation, the augmentation strategies are similar (lack fundamental differences) with [1,2,3]. For negative sampling, the so-called "reaction-aware negative sampling" is "relation-aware negative sampling", which is common in knowledge graph embedding papers. I do admit no one apply this to molecular related applications in the past.

Reproducibility: I think most of the results of the experiment part can be reproduced. But I have some problem of experiment part (see Concerns).




**Strength And Weaknesses:**

Pros:

- The paper is easy to follow.

- The three problems the paper given are important for the chemical reaction-aware self-supervised methods and hard to completely solve.

- The paper uses the chemical reaction data differently compared to existing works and get good experiment results on various tasks.

Cons:

- The paper does not have a related work part to orderly introduce the existing works and some related works mentioned in the paper are not well introduced. For example, the four GNN models used in experiment part are not introduced in the paper.

- Although the paper has experiments for various datasets, in some tasks such as chemical reaction prediction, reaction classification and yield prediction, there is only one dataset used. I think more datasets should be used to verify the efficiency of the proposed framework.

Concerns:

- In the example in Section 2.1, I cannot understand why one reaction can be transformed into two triplets, as there should be simultaneously two input molecules to get the output molecules.

- In Triplet-level Contrastive Learning in section 2.2.1, the molecules obtained after the drop step might not exist. Is it meaningful to predict these molecules that do not exist in real world?

- In table 1 in section 3.2, the results except MR are stable and almost all the stds are zero. Why?

- In table 2 in section 3.3, the results of DRFP and RXNFP are bad, what is the reason? Are there baselines that can perform better?

- The baselines for MPP tasks are too old. Are there any newer baselines? For example, some NLP based methods are mentioned in introduction. Can these methods applied in MPP tasks?


**Summary Of The Paper:**

The paper mainly proposes a chemical synthetic knowledge graph (ReaKE) to handle with different downstream tasks relevant to molecules. The proposed framework mainly addresses three problems: abnormal energy flow, ambiguous embeddings and sparse embedding space. Experimental part shows that the ReaKE improves the results in various tasks.

**Summary Of The Review:**

The paper proposes chemical knowledge graph ReaKE to learn better for different downstream tasks.
The method consists of several points which appear in existing works (contrastive learning, augmentation and negative sampling), but the combination can be new and has not been applied for molecular applications. I would recognize this paper as successfully adapt existing tricks to molecular related applications.

Overall, I think the paper is marginally below the acceptance threshold currently.

---

> ### Author Response · Authors · 2022-11-17
> **Clarification on the novelty of our work and clarification on experimental details.**
>
> Thanks for spending the time on our submission. Briefly, the reviewer has 2 main concerns: (1) The method part is lack novelty. (2) Some problems with experiments. By adding descriptions and responses, we hope our rebuttal can clarify your concerns.
>
> ## **Responses to the Cons**:
>
> **Response to Q1**: We also investigated many related methods before starting this work and appropriately applied various advanced techniques to our problem. The focus of our method is to **construct a chemical reaction KG** to solve **"Abnormal energy flow"** and **"Ambiguous embeddings"** problems, and design augmentation for triples to **obtain a denser chemical reaction embedding space**. Moreover, we designed a reaction-aware negative sampling strategy to avoid **same class negatives**, these improvements have not existed in previous work.
>
> ---
>
> **Response to Q2**: Thanks for your suggestion, we added "Related Work" in **section 4** and added detailed descriptions of GNNs in **Appendix A**.
>
> ---
>
> **Response to Q3**: We have also considered using more datasets as you mentioned, but the labeled reaction and yield datasets are scarce. Thus, we use more data splits, various sizes of the training set, and rich visualization to verify the efficiency of the proposed method.
>
> ---
>
> ## **Responses to the Concerns:**
>
> **Response to Q1**: About the question. This is actually a very insightful point, and we appreciate the reviewer for carefully noticing this. Below are more explanations. This is because in the training process of pre-training, we regard the process from reactants to products as a template-guided inference process, i.e., the constructed KG is more concerned about the **reachability** between molecules. For a reaction A+B->C, we can get A->C, B->C, and tell the relationship between A and B through the reaction template (relation) instead of A+B->C, thus helping us gain more information about **synthetic pathways** and improve our search for synthetic pathways and the prediction of possible products in more complex pretext tasks. Second, in the pre-training test stage and in the downstream tasks, we use two input molecules to predict the output molecules. That is: In the training stage of pre-training, we learned A->C, B->C through a template (relation), when we know both A and B, predicting the product C will be a much simpler task.
>
> ---
>
> **Response to Q2**: The point you mentioned is the advantage of our method. We can get a denser embedding space of molecules through such augmentation; thus the embedding of real molecules can be learned better. The same idea is also widely used in the image augmentation of the CV domain [1, 2].
>
> ---
>
> **Response to Q3**: First, MR is an indicator that is easy to have a large **variance**. It represents the average value of the real product ranking, and MRR represents the average value of the reciprocal of the real product ranking, which means when the ranking has extreme values, MR will have larger fluctuations than MRR. And there is also a situation in our results where most reactions are well predicted, but there exist some poorly predicted cases, leading to fluctuations in MR. (Assume we have 10000 samples and compare to the case where we all predicted correctly. If the predicted ranking for a sample becomes 10000, MRR will drop to 0.9999(-0.0001), but MR will rise to 1.9999(+0.9999)).
>
> ---
>
> **Response to Q4**: This is due to the fact that these two methods cannot learn the difference between reactants and products well compared to our method, and we also supplement the results of the most competitive MolR method. (See **Figure 3(b)**)
>
> ---
>
> **Response to Q5**: Since we focus on the most relevant reaction classification task and yield prediction task, and our baseline has demonstrated the superiority over these NLP-based methods [3], we only compare with the baseline and several traditional methods to demonstrate the effectiveness of our methods on molecular representation learning.
>
> ---
> [1] He K, Fan H, Wu Y, et al. Momentum contrast for unsupervised visual representation learning[C]//Proceedings of the IEEE/CVF conference on computer vision and pattern recognition. 2020: 9729-9738.
>
> [2] Chen T, Kornblith S, Norouzi M, et al. A simple framework for contrastive learning of visual representations[C]//International conference on machine learning. PMLR, 2020: 1597-1607.
>
> [3] Wang H, Li W, Jin X, et al. Chemical-Reaction-Aware Molecule Representation Learning[C]//International Conference on Learning Representations. 2021.

---

### Official Review · Reviewer_oZTX · 2022-11-02

**Confidence:** 4
**Correctness:** 3
**Technical Novelty And Significance:** 2
**Empirical Novelty And Significance:** 3
**Recommendation:** 5

**Clarity, Quality, Novelty And Reproducibility:**

Clarity and quality:
Generally, the clarity of this paper is unsatisfactory. As a work of interdisciplinary domains across chemistry and molecular science, graph machine learning and representation learning, many terms, and concepts mentioned in this paper may not be understood universally without formal introduction and clarifications. A notation table or preliminary section for the glossary used in this paper is highly suggested, to explain the entire KG schema as shown in Figure 2(a) in Section 2.1 (Section 2.1 is insufficient to understand the full image). To name a few unclarified terms:
1. In-depth analysis of the disadvantages of existing methods. All three challenges such as energy flow, ambiguous embedding, sparse embedding space (and smoothness) are not well justified in the introduction and how are these problems observed or demonstrated?
2. What does it mean by "embeddings are equal in embedding space"?
3. Template information, reaction condition, and how it is used and guides the model learning
4. Full node features in molecular encoder as one-hot input features (how degree information is included as one-hot)
5. Why are the design of triple-level and graph-level contrastive learning / negative sampling valid? Why is the proposed method for hard negative examples more informative, compared to the original negative sampling?
6. All the tasks are not explicitly explained about the input, output, and goals and how the inference steps are done.

Novelty:
This work is considered relatively innovative (or marginally innovative) as a straightforward combination of well-known modules as building blocks. Most of the techniques used in this paper, are molecule representations (GNN, graph readout), self-supervised learning, augmentation, and contrastive learning, and knowledge graph embedding (TransE-like structure). It is beneficial to have a comprehensive set of technical approaches that collectively work with the input of relation-level graphs and molecule-level graphs, as an integrated and improved solution.

Reproducibility:
The level of reproducibility is medium. ReaKE involves many modules as building blocks (GNN, SSL, mini-batch negative sampling, joint training), however, it seems that many implementation details and hyperparameters are not fully mentioned, including visualization of tSNE configuration. The supplementary materials may cover some model specifications but without a pre-released codebase, it is not confident to reproduce the results. Datasets are publicly available research datasets.

Please check the section "summary of the review" for more detailed questions, comments, and suggestions.

**Details Of Ethics Concerns:**

Not applicable.

**Strength And Weaknesses:**

Strength:
1. Well-established problem and formulation of chemical synthesis problem from the perspective of "multi-view" KG representation learning
2. The proposed framework is well-designed and mostly technically sound.
3. Extensive experiments from multiple applications and comparison with a large group of the state-of-the-art baseline approaches

Weaknesses:
1. The writing clarity needs improvement, especially on the term definitions, and specific mathematical functions.
2. Insufficient justifications for the rationale of the proposed model and training design.
2. Some important baselines are not included in the comparison, discussion, or related work. It is unclear to distinguish previous work on graph learning and SMILE-based models.


**Summary Of The Paper:**

This paper proposed a new framework, named ReaKE for learning representations for chemicals to better predict chemical reactions
The main contributions of this work are three folds as claimed by the authors: (1) chemical synthesis KG of reactants, products and; (2) contrastive learning strategies in KG triple level and molecular graph level; (3) joint training on reaction and molecular level which are essentially two types of graph.
Experiments have demonstrated that ReaKE outperforms other SOTA models and the effectiveness of ReaKE modules with ablation studies.


**Summary Of The Review:**

This paper proposed ReaKE as a comprehensive framework for chemical-KG representation learning both from the reaction knowledge graph and the molecular atom graph. The major issue of this paper is clarity on model details which makes it hard to fully understand the motivation and rationale of the learning strategies and training protocol.

Some other questions and suggestions are as follows:
1. There is no related work in this paper which brings difficulties to understand the position of the paper. It is suggested that a separate section discuss the distinction and how the baseline models are selected. Some works are missing regarding SSL in graph neural networks [1], graph-sequence hybrid models [2], and multi-view KG joint training [3].
2. Why are traditional KG embedding methods on reaction KG (TransE) not usable for comparison for downstream tasks?
3. No report of scalability and computing resources used for the complex learning framework.
4. As a minor issue, the term "synthetic“ may cause confusion at first when it comes to "synthetic KG" which by default means manually generated datasets instead of real-world ones. "Chemical Synthesis Knowledge Graph (CSKG)" may be used to avoid misunderstanding.


References:
1. Xie, Y., Xu, Z., Zhang, J., Wang, Z., & Ji, S. (2022). Self-supervised learning of graph neural networks: A unified review. IEEE Transactions on Pattern Analysis and Machine Intelligence.
2. Wang, Z., Liu, M., Luo, Y., Xu, Z., Xie, Y., Wang, L., ... & Ji, S. (2022). Advanced graph and sequence neural networks for molecular property prediction and drug discovery. Bioinformatics, 38(9), 2579-2586.
3. Hao, J., Ju, C. J. T., Chen, M., Sun, Y., Zaniolo, C., & Wang, W. (2020, September). Bio-JOIE: Joint representation learning of biological knowledge bases. In Proceedings of the 11th ACM International Conference on Bioinformatics, Computational Biology and Health Informatics (pp. 1-10).

For other issues such as clarity and novelty, please check the corresponding sections.

---

> ### Author Response · Authors · 2022-11-17
> **Clarification on the main motivations and clarification on downstream tasks.**
>
> ## **Responses to the clarification problems.**
> **Q1: In-depth analysis of the disadvantages of existing methods. All three challenges are not well justified in the introduction and how are these problems observed or demonstrated?**
>
> **Q2: What does it mean by "embeddings are equal in embedding space"?**
>
> **Response**: First, we emphasize the motivation of our work as a response for Q1 and Q2.
>
> (1) we think that there are some misunderstandings about the motivation of our method since we did not explain the baseline clearly enough. First, the main assumption in the baseline mentioned in Q2 is that for a given chemical reaction CH3COOH +C2H5OH->CH3COOC2H5+H2O, it assumes e(CH3COOH)+e(C2H5OH)=e(CH3COOC2H5)+e(H2O), e(·) represents the molecule embedding function. From this assumption we can see that the **"Abnormal energy flow"** problem is macroscopic exists, because two molecules can flow between each other as long as their embeddings are equal. First, this violates the entropy change principle in chemical reactions. Second, this will lead to some errors in modeling reactions (see the detailed example in our article).
>
> (2) For the **"Ambiguous embeddings"** problem, we can see the embeddings of the product and the reactant are close under the assumption, this will lead to a problem that the product is predicted as the reactant (see **No.5** and **No.72** reactions in Table 5).
>
> (3) For the **"Sparse embedding space"** problem, it's because chemical reactions are limited, that is the chemical space is sparse which will make the learned embeddings prone to large deviations under perturbation. Therefore, we make more reactions through data augmentation to expand our data and make the embedding space denser.
>
> ---
>
> **Q3: Template information, reaction condition, and how it is used and guides the model learning.**
>
> **Response**: Briefly speaking, the template information is involved as the **relation** of triplets in CSKG, it helps establish relationships between molecules. The flow between reactants and products needs to be triggered by environmental conditions such as temperature and pressure, and the reaction conditions are finally reflected on the reaction site which can be represented as the reaction template. we added more description in the revised manuscript, see **section 3.1** and **3.2.2** for more details.
>
> ---
>
> **Q4: Full node features in molecular encoder as one-hot input features (how degree information is included as one-hot).**
>
> **Response**: We totally have atom feature of **69-dimension**, and we also add this detail in **section 2.2.1**.
>
> ---
>
> **Q5: Why are the design of triple-level and graph-level contrastive learning / negative sampling valid? Why is the proposed method for hard negative examples more informative, compared to the original negative sampling?**
>
> **Response**: Briefly speaking, the main purpose of triplet-level learning is to **construct a relatively dense and smooth embedding space for triplets** as we generate more possible triplets. While the graph-level learning aims at **improving the model's ability of distinguishing noise samples** and making the training more effective. And compared to the original negative sampling, we design reaction-aware sampling strategy to avoid **same class negatives**. We added more clearly descriptions in **section 2**.
>
> ---
>
> **Q6: All the tasks are not explicitly explained about the input, output, and goals and how the inference steps are done.**
>
> **Response**: Thanks for the suggestion, we added input and output descriptions and more experimental details in the revised manuscript.
>
>
> ---
>
> ## **Reproducibility**:
> **Response**: We have uploaded our code at the supplementary material.
>
> ---
>
> ## **Other questions**:
> **Response to Q1**: Thank you for your suggestion, we added "Related Work" in **section 4** to help understand the position of our work and the "how the baseline models are selected" have been stated in the **"baseline"** part of every task. Most of the baseline selections are based on the previous correlated literature.
>
> ---
>
> **Response to Q2**: The core of our work is to demonstrate that **the embeddings we extracted from the reactions are useful for the pre-trained GNN** and **transferable to downstream reaction-relevant tasks**. While in the downstream reaction classification and yield prediction tasks, our main purpose is to evaluate the quality of the reaction embeddings. Other KG embeddings methods can be easily replaced in the framework but do not help to elucidate this purpose.
>
> ---
>
> **Response to Q3**: (1) "scalability problem": we cannot try more pre-training datasets due to the limited chemical reaction dataset at present, and if there is a larger chemical reaction dataset in the future, we will also try it in our framework. (2) The “computing resources” is added in **Appendix C**.
>
> ---
>
> **Response to Q4**: Good point, we have used "Chemical Synthesis Knowledge Graph (CSKG)" in the article accordingly.

---

### Decision · Program_Chairs · 2023-01-20

**Decision:**

Reject

**Justification For Why Not Higher Score:**

Limited novelty comparing MoIR. Experiment part is unclear.

**Justification For Why Not Lower Score:**

N/A

**Metareview: Summary, Strengths And Weaknesses:**

This work proposes a molecular representation learning method based on pre-training on a chemical synthetic knowledge graph.
The proposed framework mainly addresses three problems: abnormal energy flow, ambiguous embeddings and sparse embedding space. Experimental part shows that the ReaKE improves the results in various tasks.

Strength of the paper:
1. The paper has novel contribution by considering the abnormal energy flow, and ambiguous embedding.
2. The contrastive loss using knowledge graph.

Weakness of the paper:
1. Limited technical advancement comparing to existing work MoIR.
2. The experiments lack adequate description. Much of the setup is unclear and hard to reproduce in current description.

In summary, the paper is not ready to publish in the conference yet.

**Summary Of Ac-Reviewer Meeting:**

We discussed the main contribution of the paper (abnormal energy flow, ambiguous embeddings) and its main limitation (novelty comparing to MoIR, experimental description).